# Data Fusion Approach to Simultaneously Evaluate the Degradation Process Caused by Ozone and Humidity on Modern Paint Materials

**DOI:** 10.3390/polym14091787

**Published:** 2022-04-27

**Authors:** Laura Pagnin, Rosalba Calvini, Katja Sterflinger, Francesca Caterina Izzo

**Affiliations:** 1Institute of Science and Technology in Art, Academy of Fine Arts Vienna, Schillerplatz 3, 1010 Vienna, Austria; k.sterflinger@akbild.ac.at; 2Department of Life Sciences, University of Modena and Reggio Emilia, Via Amendola 2, 42122 Reggio Emilia, Italy; rosalba.calvini@unimore.it; 3Department of Environmental Sciences, Informatics and Statistics, Ca’ Foscari University of Venice, Via Torino 155/b, 30174 Venice, Italy; fra.izzo@unive.it

**Keywords:** data fusion, acrylic paints, styrene-acrylic paints, ozone aging, FTIR, PY-GC/MS

## Abstract

The knowledge of the atmospheric degradation reactions affecting the stability of modern materials is still of current interest. In fact, environmental parameters, such as relative humidity (RH), temperature, and pollutant agents, often fluctuate due to natural or anthropogenic climatic changes. This study focuses on evaluating analytical and statistical strategies to investigate the degradation processes of acrylic and styrene-acrylic paints after exposure to ozone (O_3_) and RH. A first comparison of FTIR and Py-GC/MS results allowed to obtain qualitative information on the degradation products and the influence of the pigments on the paints’ stability. The combination of these results represents a significant potential for the use of data fusion methods. Specifically, the datasets obtained by FTIR and Py-GC/MS were combined using a low-level data fusion approach and subsequently processed by principal component analysis (PCA). It allowed to evaluate the different chemical impact of the variables for the characterization of unaged and aged samples, understanding which paint is more prone to ozone degradation, and which aging variables most compromise their stability. The advantage of this method consists in simultaneously evaluating all the FTIR and Py-GC/MS variables and describing common degradation patterns. From these combined results, specific information was obtained for further suitable conservation practices for modern and contemporary painted films.

## 1. Introduction

Since their experimentation in the 1950s and the subsequent industrial development in the 1980s, acrylic and styrene-acrylic materials have had an important influence on the evolution and use of polymeric binders used in the artistic field [1]. In fact, their chemical-physical properties, such as high versatility, miscibility, covering power, and fast-drying, have made these materials extremely popular, resulting in the presence of numerous artworks made with these binders and, nowadays, present in the most important contemporary art collections [2,3]. These artworks are generally exhibited both outdoors (such as murals, polychrome statues, etc.) and in indoor museum environments. If, in the first case, it is clear how pollutant climatic agents (such as UV-light, temperature, humidity, corrosive gases) affect the physical-chemical stability of the painted materials in an uncontrolled way, in indoor conditions, they should be better evaluated and monitored. Unfortunately, this is not always the case. Specifically, ozone (O_3_) is the most present gaseous pollutant in outside and inside environments [4]. If outdoor, it is a product deriving from fossil fuel combustion processes and the consequent atmospheric photochemistry reactions [5,6], in museum exhibitions, it can be present due to old ventilation systems, ozone-based restoration treatments [7], incorrect conservation practices in archives and storages [8], and the consequent inadequate monitoring of its concentration levels [9,10,11]. These factors, combined with the presence of other degrading agents, are precursors to degradation reactions that affect artistic materials. Their understanding still remains a topic of current interest as, especially for modern and contemporary works of art, they are more complex due to the presence of multiple constituents, different formulations of the materials that change over the years, and the presence of new materials not yet identified. As many research studies state [12,13,14,15], modern polymeric paint materials, such as spray graffiti, show different degradation behaviours depending on their chemical composition. The composition of pigments, extenders, fillers, and additives can influence the overall stability of the organic matrix [16,17], especially when exposed to the main weathering agents, such as light radiation [18,19]. However, degradation effects related to the action of gaseous corrosive agents (even in a humidified environment) have been considered only in limited research [20,21]. This paper presents the study of the ozone influence on the degradation of acrylic and styrene-acrylic binders. Accelerated aging experiments were performed, combining the O_3_ with two different humidity contents (50 and 80 RH%) for a total exposure period of 168 h. The samples examined were paint films obtained by mixing the above-mentioned synthetic binders with nine inorganic pigments. The latter were selected because their well-known and documented use in paint production is still present in recent formulations [22,23,24,25].

The main objective of the study was to understand the chemical degradation processes that occurred on the samples. Attenuated total reflection-infrared spectroscopy (ATR-FTIR) and pyrolysis-gas chromatography/mass spectrometry (Py-GC/MS) were chosen. From the evaluation of the data obtained, it was possible to characterize the main functional groups of the binders, both before and after aging, understand which degradation reactions occur after the exposure to ozone, and differentiate the observed behaviour according to the presence of various pigments, also carrying out a semi-quantitative evaluation. It has to be highlighted that ATR-FTIR and Py-GC/MS techniques allow to evaluate different aspects of the degradation process occurring on the paint samples. While ATR-FTIR measures information related to the surface of the samples, Py-GC/MS analyses the bulk modifications occurring not only on the sample’s surface, but also in deeper layers. With these considerations, it is necessary to specify that this study will not extensively investigate the chemical-physical behavior of the additives present in the binders when exposed to accelerated aging. In fact, the information related to the degradation processes of polymeric binders initiated by ozone exposure are still unclear and this study aims to expand the knowledge of this issue. To better investigate this aspect, and to obtain comparable data between the two analytical techniques suitable for the following statistical evaluation, specific experimental conditions (specifically, Py-GC/MS) were selected. However, they did not make it possible to obtain information on the contribution of additives in the degradation processes. The latter interesting and more detailed research topic could be extended in future studies by combining, e.g., the results of other surface or bulk techniques. As mentioned, the information obtained in this study from both ATR-FTIR and Py-GC/MS was integrated using data fusion techniques.

Basically, it consists in jointly analyzing the data blocks resulting from different analytical techniques to provide a more comprehensive characterization of the system under investigation [26]. Data fusion is gaining an increasing interest thanks to the significant advantages arising from the possibility of jointly analyzing the considered samples using different analytical techniques to increase the quantity and the quality of the information that can be extracted. Due to the nature and dimensionality of the data obtained by several analytical techniques, it is necessary to examine the merged dataset using a multivariate statistical approach in order to retrieve useful information related to common patterns existing in the considered data blocks as well as existing relationships among variables acquired from different instruments [27]. Several research studies highlighted the advantages of a data fusion approach in different research fields, including metabolomics [28,29,30], pharmaceutical analysis [31,32], food safety, and quality control [33,34,35]. While in the research mentioned above, the application of data fusion techniques can be considered an established tool, it still has limited applications in cultural heritage studies. Considering the characterization of art materials, previous studies combined different spectroscopic techniques, such as Raman spectroscopy, X-ray fluorescence, visible-near infrared spectroscopy, or FTIR spectroscopy, to classify pigments of different paint layers [36,37,38]. Data fusion can be applied at three levels: low-level, mid-level, and high level [39,40]. Low-level data fusion is the simplest and most straightforward method for jointly evaluating multiple data blocks, and it essentially consists in simply merging the different datasets row-wise [41]. Mid-level data fusion is based on the concatenation of relevant features selected or extracted from the considered data blocks. The datasets obtained from the different analytical techniques are separately analyzed to extract relevant features based on variable sections or data reduction methods [42]. Finally, in high-level data fusion, the outcomes of different models individually calculated on the single data blocks are merged together to provide the final output [43].

In the current experiment, the two datasets containing the ATR-FTIR spectra and the compounds identified by Py-GC/MS were merged at the low-level, and the fused matrix was then analyzed using principal component analysis (PCA) in order to comprehensively characterize the degradation phenomena occurring on samples due to the exposure to ozone and humidity, and identify common degradation trends observable both on the sample surface and in the paint bulk.

## 2. Materials and Methods

### 2.1. Sample Preparation

Forty mock-ups were prepared by mixing pure acrylic emulsion Plextol^®^ D498 (Kremer Pigmente, Aichstetten, Germany) and styrene-acrylic binder Acronal S790 (BASF, Ludwigshafen, Germany) with nine different inorganic pigments (Kremer Pigmente, Aichstetten, Germany). To allow a reproducible investigation of the degradation behavior, homogeneous and consistent mixtures (similar to the commercial formulations) were created, with a pigment/binding medium (P/BM) ratio of 1:3 in weight. The mock-ups were prepared by mixing the pigment and binding medium in a ceramic mortar and casting the obtained paint on glass slides. The preparation method used is the so-called doctor-blade technique [44], allowing to reach a film thickness of 150 μm. The samples were dried at ambient conditions (approximately 22 °C and 30% RH) for three weeks. The pigments and binders considered in this study are listed in Table 1.

### 2.2. Weathering Experiments

The aging chamber used (Bel-Art™SP Scienceware™, Vienna, Austria) is made of a co-polyester glass (Purastar^®^, Hanoi, Vietnam), which allows to reach a total volume of 30 cm^3^ and is equipped with gas inlets and outlets. An aging system is connected to the chamber to mix synthetic air with corrosive gases. Synthetic air 5.0 (Messer, Vienna, Austria) is initially humidified using double distilled water and then mixed with the selected gas, in this case, ozone. During aging, the chamber is constantly flushed with the gas mixture with a gas flow rate of 100 L/h. The aging experiments on styrene-acrylic paints used a relative humidity (RH) content of 50% and 80% and an ozone concentration (O_3_) of 2500 ppm for a total exposure of 168 h. The gas concentration was periodically monitored during artificial aging using a specific sensor for detecting O_3_ (Aeroqual Limited, Auckland, New Zealand, model AQL S200). The ozone value in the chamber may vary by ±10–13 ppm. To reproduce outdoor accelerated aging, the value of the annual average concentration in the atmosphere (monitored by the European Environment Agency for Air Quality Monitoring) was considered [45]. It is approximately 40 ppm. Therefore, considering this value and the experimental concentration chosen, the exposure time deriving from the accelerated aging carried out is approximately equal to 1 year.

### 2.3. Attenuated Total Reflection Fourier Transform-Infrared Spectroscopy (ATR-FTIR)

IR measurements were performed using a LUMOS FT-IR microscope (Bruker Optics, Ettlingen, Germany), equipped with a cooled photoconductor MCT detector. It was used in ATR mode in combination with a germanium crystal. The latter, having a refractive index (n_1_) of 4.01 and allowing to reach an angle of incidence (θ) of the IR beam of 45°, is able to obtain information in depth around 0.65 μm. For each sample, five measurement points were acquired in a spectral range between 4000 and 480 cm^−1^ (64 scans with a resolution of 4 cm^−1^). The resulting spectra were collected and evaluated by OPUS^®^ 8.0 software (Bruker Optics, Ettlingen, Germany). Subsequently, the spectra were pre-processed by averaging, baseline correction, and vector normalization. From the treated spectra, the semi-quantification of specific absorption bands was performed. As shown in a previous study [46], the bands to be integrated are chosen because they show a more significant spectral change after exposure to pollutants and represent more reliable data for this evaluation. Considering that according to the different polymeric binder and aging conditions (RH%) the degradation products are different, specific functional bands were chosen for the integration. Specifically, the bands at 1703 and 1650 cm^−1^ were selected for styrene-acrylic paints, and at 1650 and 1343 cm^−1^ for acrylic ones. More detailed information is reported in the following sections. In the specific case of styrene-acrylic paints, the Fourier self-deconvolution function was used too. It is able to obtain well-resolved peaks from previously convoluted bands (broadened) by the same type of line-broadening function. In general, deconvolution is equivalent to a multiplication of the interferogram. In this case, the deconvolution function for Lorentzian shapes was used, with a deconvolution factor of 13 and a noise reduction factor of 0.63 (half interferogram length).

### 2.4. Pyrolysis-Gas Chromatography/Mass Spectrometry (Py-GC/MS)

Small fragments of acrylic and styrene-acrylic samples were accurately weighed using a microanalytical balance (0.001 mg), placed in eco-cup pyrolysis crucibles, and investigated with Single Shot Py-GC/MS (SS-Py-GC/MS). A PY-3030D pyrolyzer (Frontier Lab., Koriyama, Japan) was used, mounted through programmed temperature vaporization (PTV) injector on a Trace 1310 gas chromatograph (ThermoFisher Scientific, Waltham, MA, USA), and combined with an ISQ7000 mass spectrometer (ThermoFisher Scientific, Waltham, MA, USA). Based on previous experimental set-ups [21,47], the SS-Py-GC/MS analyses were performed at 600 °C for 0.20 min. Separation took place on a Supelco SLB5 MS column (20 m, 0.18 mm, 0.18 μm). Helium with a constant flow of 0.9 mL/min was used as the carrier gas. The temperature program used was: 35 °C (held 1 min)–16 °C/min−220 °C–10 °C/min−315 °C (held 2 min). The interface temperature was 280 °C, and the ion source was 300 °C. Mass spectra were recorded between 20 and 650 *m*/*z* with 0.2 sec of dwell. Chromeleon software was used for collecting and processing the mass spectral data. The qualitative interpretation of the results was achieved with NIST Libraries, F-Search software, and Automated Mass spectral Deconvolution and Identification System (AMDIS) software ad hoc created libraries. Furthermore, the ESCAPE system, an expert system for characterizing Py-GC/MS data using AMDIS & Excel, was used [48]. The data were further processed in order to obtain a semi-quantitative evaluation. Through Chromeleon software, two specific processing methods were created for acrylic and styrene-acrylic paints. All peaks listed in Table 2 and Table 3 were integrated, and the relative areas were then corrected and normalized considering the different weights of the samples (which generally varied between 70 and 80 µg).

### 2.5. Data Organization for Data Fusion

Low-level data fusion and multivariate data analysis were carried out, considering separately the samples prepared with the two different binders. For each binder type, before applying low-level data fusion, the two datasets resulting from ATR-FTIR spectroscopy and Py-GC/MS chromatography were separately pre-processed. Regarding the FTIR spectra, only the spectral region between 3030 cm^−1^ and 1500 cm^−1^ was considered for data fusion in order to account only for the spectral response ascribable to the binder signals. Subsequently, the spectra were mean-centered. On the other hand, autoscaling was applied to the dataset containing the compounds identified by Py-GC/MS. The resulting fused datasets were composed of a total of 760 variables for the ATR-FTIR data, and 26 and 41 Py-GC/MS variables for acrylics and styrene-acrylics samples, respectively (Table A1 and Table A2). Since the FTIR data block has a much higher number of variables than the Py-GC/MS data block, it was necessary to apply an additional pre-processing step to the fused datasets to avoid the risk that the block strongly influences the results with the highest number of variables. For this reason, the fused datasets were pre-processed using block scaling, which consists in scaling each data block according to its global standard deviation. In this way, the two data blocks will contribute with equal weights to the subsequent calculation of multivariate models, while the relative weights of the variables within each block are preserved. The fused datasets were then analyzed using principal component analysis (PCA) to evaluate the degradation patterns due to ozone and relative humidity exposure. Data fusion and PCA modeling were performed using the PLS_Toolbox software (v. 8.8.1, Eigenvector Research Inc., Wenatchee, WA, USA) working under a MATLAB environment (R2020a, The MathWorks, Natick, MA, USA). Figure 1 shows a schematic representation of the workflow followed in this study for the data fusion of ATR-FTIR and Py-GC/MS data blocks.

## 3. Results and Discussion

### 3.1. Identification and Degradation Investigations

#### 3.1.1. ATR-FTIR Results of Styrene-Acrylics

In Figure 2, the main functional groups of the styrene-acrylic polymer can be identified. The functional groups belonging to the acrylate-based fraction and phenyl group can be distinguished. The acrylate component can be characterized by the C–H bond stretching vibrations (at 2956–2930–2872 cm^−1^), the C=O stretching (at 1721 cm^−1^), and the additional bands in the fingerprint region of the C–C and C–O stretching (at 1154–1128–1066 cm^−1^). The bands of the C–H stretching (at 3083–3061–3027 cm^−1^), the C=C stretching (at 1601 cm^−1^), the C–C vibration (at 1493–1454 cm^−1^), and the C–H bending (at 759–698 cm^−1^) are observed, related to the aromatic ring of the phenyl group [49].

As shown in Figure 2, two different degradation trends are observed, depending on the amount of relative humidity used during accelerated aging. At 80% RH, the main functional groups of the styrene-acrylic film did not significantly change. The two trends are the most evident factors of the degradation process: the increase of the intensity of almost all the spectral signals due to the strong hydrolytic degradation, and the decrease of the band at 759–698 cm^−1^ belonging to the aromatic CH out-of-plane bending of the phenyl group. In fact, ozone initially attacks the aliphatic double bonds, the aromatic double bonds, or the saturated aliphatic chains, thus leading to the decrease of the phenyl group signal and the growth of a shoulder around 1665 cm^−1^, relative to the formation of carbonyls and carboxylic acids [50]. In contrast, at lower amounts of relative humidity (50% RH), the styrene-acrylic polymeric film appears to undergo the most significant degradation. In Figure 2c, the spectrum obtained shows that the spectral signals belonging to the phenyl component of the molecule are no longer detectable. In addition, it can be seen that the band at 1727 cm^−1^ appears to shift and become broader. After pre-treatment of the spectra and subsequent deconvolution of the band under investigation, it was possible to observe that the band is not shifted but covered by the band’s formation at 1703 cm^−1^. With the deconvolution, it was also possible to characterize the presence of two small peaks at 1670 and 1650 cm^−1^. These spectral responses obtained from accelerated aging suggest the possible decomposition of the phenyl group with consequent cleavage of the double bonds, forming species, such as aldehydes, ketones, and carboxylic acids [51,52,53,54]. A further demonstration of the polymer degradation by chain scission is given by the increase in absorption around 3500 cm^−1^ related to the formation of carbonyls and carboxylic acids [55]. As already demonstrated in a parallel study [56], this phenomenon is explained by the intrinsic properties of the water deposited on the polymeric surface and the ozone reactivity in a humid environment. Depending on the humidity value used, the layers of superficial water can deposit differently. In fact, the water will be arranged in a double layer system at 50% RH. Therefore, the degradation products deriving from the water/ozone combination will be more concentrated and more in contact with the polymeric surface. On the contrary, with relative humidity values of 80%, the accumulated water monolayer will be more extensive and thicker, reducing the intermediate reactions of the ozone in the water film and the consequent corrosive action [57,58].

#### 3.1.2. ATR-FTIR Results of Acrylics

Through ATR-FTIR analysis, it was possible to identify the main functional bands of the acrylic films. It was characterized as nBA/MMA as it shows C–H bond stretching vibrations (at 2956–2875 cm^−1^), C=O stretching (at 1726 cm^−1^), CH_2_- and -CH_3_ bending vibrations (1452 and 1386 cm^−1^, respectively), and the C–O–C and C–O stretching bands (at 1236, 1160, 1146 cm^−1^) in the fingerprint region [46,59,60]. Furthermore, the spectral signal of a surfactant, namely PEO (polyethylene oxide), was identified by the three characteristic absorbance bands at 2891, 1343, and 1112 cm^−1^ [61,62].

In Figure 3, it is possible to observe that the main spectral change is associated with the increase in the intensity of the acrylic functional groups for all the samples aged with both 50% and 80% RH. This degradation phenomenon is related to the hydrolysis reactions caused by the action of humidified water on the polymeric film, causing the gradual opening of the polymeric molecular structure. However, some spectral differences are observed between the two sets of aged samples. As discussed in the IR results of styrene-acrylic paints, the functional groups of the acrylic film do not undergo a noticeable change at 80% RH. The only degradation factor shown is the intensity increase in the surfactant functional bands. As reported in several studies [61,63,64,65], PEO is a particularly hygroscopic non-ionic surfactant that favours the absorption of water by the polymeric material in which it is mixed and subsequently tends to migrate into the air–film interface. It is one of the most important degradation processes of acrylic paints since it affects the physical-mechanical resistance, the adhesion of the film to the substrate, the superficial aesthetical properties, and favours the accumulation of particulate matter, dirt, and the interaction with other deteriorating agents [66].

From the comparison with a study carried out by exposing modern paints to the degrading action of SO_2_ and NO_x_ [67], the migration of the surfactant is observed in all three experiments, confirming that the acrylic emulsions are subject to degradation deriving from high concentrations of humidity, as well as from the different gas pollutants (NO_x_ and O_3_ favor the migration of PEO more than SO_2_). However, this migration process for acrylic emulsions aged with SO_2_ and NO_x_ also occurs at 50% RH, while with O_3_, the absorption bands of PEO are not detected. Furthermore, degradation reactions affecting the acrylic component are observed. The carboxyl C=O band broadens, its intensity increases compared to the unaged sample, and a peak at 1650 cm^−1^ is detected (also observed for styrene-acrylic paints) related to the formation of species, such as aldehydes, ketones, and carboxylic acids. As previously discussed, the chemical properties of humidified water at 50% in combination with the degrading power of O_3_ cause oxidizing effects on polymeric emulsions greater than 80% RH. Therefore, it promotes the breaking of the aliphatic and carboxylic bonds of the acrylic component. Assuming that even at 50% RH the migration of the surfactant has occurred as with the other gaseous agents, it is hypothesized that the O_3_ in solution leads to the complete dissolution of the PEO particles; for this reason, the surfactant IR signals are no longer detected.

#### 3.1.3. Py-GC/MS Results of Styrene-Acrylics

Table 2 reports the Py-GC/MS analysis results of pure unaged styrene-acrylic emulsion. The commercial product, called Acronal^®^ S790 (BASF, Ludwigshafen, Germany), was identified by intense peaks mainly related to the styrene component and, in many cases, also acrylic ones. According to the literature [68,69], the most commonly acrylic copolymers added to these formulations are p(EA−MMA), p(nBA−MMA), and p(2EHA−MMA). In terms of chemical stability, this combination determines a glass transition temperature (Tg) value of the dried film high enough to prevent the binder from becoming tacky and low enough to avoid cracking. From the observation of the chromatogram in Figure 4, the styrene-acrylic emulsion shows a combination of BA (*m*/*z* = 55, 73) at RT 3.37 min, nBMA (*m*/*z* = 69, 41, 87, 56) at RT 3.88 min, styrene (*m*/*z* = 104, 78) at RT 3.33 min, and α-methylstyrene (*m*/*z* = 118, 103, 78) at RT 3.93 min. Although the latter compound derives from the pyrolysis of polystyrene [70], its presence in large quantities suggests that the α-methyl-styrene monomer was added individually within the final polymeric formulation. From the identification of nBMA, the presence of other compounds due to the pyrolytic fragmentation of the monomer is observed, such as 2-butene (*m*/*z* = 41, 56) at RT 0.97 min, butyl alcohol (*m*/*z* = 56, 31) at RT 1.79 min, and the BA trimer (*m*/*z* = 41, 57, 134) at RT 13.84 min. Additional peaks related to the combination and fragmentation in the pyrolysis were detected, such as BA-styrene dimers (*m*/*z* = 91, 115, 104, 130, respectively), styrene-styrene dimer (*m*/*z* = 142, 129, 115, respectively), and BA-BA-styrene trimers (*m*/*z* = 98, 91, 126, respectively). It is interesting to note that ethylbenzene, naphthalene, and some of its monomers (such as dihydro- and 1-methyl-naphthalene) were identified at RT 3.13, 5.26, 5.45, and 6.25 min. The presence of benzene derivatives and polycyclic aromatic hydrocarbons may be associated with the industrial employment of organic solvents for the formulation of new polymeric products, the presence of which could lead to VOCs exposure [71].

Comparing the results obtained from pure unaged and aged styrene-acrylic samples reveals that the effects of ozone exposure vary, depending on the RH% values used. Observing the intensity values of the peak areas of the three sets of samples (Figure A1), it is possible to notice that those aged at 50% RH show the most significant degradation effects. In fact, peaks, such as styrene, styrene dimer 4, styrene trimer 2, styrene-styrene-nBA trimer, nBA-styrene dimer 2, and nBA-nBA styrene trimer 1 and 3, have a higher intensity than in unaged samples. This behavior highlights that the high concentration of ozone, mixed with different RH amounts and deposited on the sample’s surface, induced polymeric cleavage reactions mainly of the styrene groups and, partially, of the butyl acrylate ones linked to the phenyl ring. On the other hand, samples aged at 80% RH show no particular degradation trends. Compared to the unaged samples, almost all the polymeric fractions increase their intensity of the peaks area, indicating that hydrolysis reactions are taking place. The trend of the α-methylstyrene fraction is also interesting. In fact, it is the only peak that at 80% RH value that increases considerably more than the unaged and 50% RH aged samples. α-methylstyrene can be degraded by oxidation with OH radicals and ozone, although the interaction with OH radicals mainly stimulates the degradation processes of this polymeric fraction [72].

#### 3.1.4. Py-GC/MS Results of Acrylics

Table 3 reports the Py-GC/MS analysis results of pure unaged acrylic emulsion. The commercial product, called Plextol^®^ D498 (Kremer Pigmente, Aichstetten, Germany), was identified as a co-emulsion of poly(butyl acrylate/methyl methacrylate), p(nBA/MMA). This co-polymer was the most used in the artists’ paint formulations due to its high physical-chemical properties [73]. In Figure 5, the chromatogram of pure unaged acrylic emulsion depicted the most intense peak of MMA (*m*/*z* = 69, 41, 100) at RT 2.07 min, and nBMA (*m*/*z* = 69, 41, 87, 56) at RT 3.89 min. Additionally, peaks corresponding to acrylic acid methyl ester (*m*/*z* = 55, 27, 85) at RT 1.48 min and butyl alcohol (*m*/*z* = 56, 31) at RT 1.79 min were detected. The pyrolysis of BA (*m*/*z* = 55, 73) at RT 3.37 min is quite complex, and different fragments were formed, such as 2-butene, n-butyl aldehyde, n-butyl acetate, butyl acrylate, and n-butyl tiglate [74]. At higher retention times, several peaks were recorded, indicating the presence of dimeric and trimeric fractions. In fact, by observing the corresponding pyrograms, it is possible to identify the nBA sesquimer (*m*/*z* = 115, 87, 171) and nBA dimer (*m*/*z* = 127, 126, 98) at RT 9.07 min and at RT 9.41 min, respectively. Moreover, two peaks at RT 7.30 min and at RT 7.53 min were identified as nBA-MMA sesquimer (*m*/*z* = 143, 129, 83) and nBA-MMA dimer (*m*/*z* = 112, 67, 95). Finally, the nBA trimer (*m*/*z* = 181, 134, 236) at RT 13.86 min and the nBA-nBA-MMA trimers (*m*/*z* = 195, 93, 194) at RT 12.06, 12.22, and 12.85 min were detected.

The comparison between the results obtained from pure unaged and aged acrylic samples reveals that the effects of exposure to ozone are different, according to the RH% values used. For the samples aged at 50% RH, an increase in the area value of the main peaks, i.e., butyl alcohol, MMA, BA, and nBMA, is observed (Figure A2). In fact, as seen in the FTIR results, these aging conditions promote the hydrolysis reactions of the ester groups of the acrylic polymer and favour the area increase of the compounds previously mentioned. On the other hand, with a higher relative humidity value (80%), acrylic acid and propanoic acid areas increase, while those of butyl alcohol and n-butyl acetate decrease. This demonstrates that, unlike the FTIR results, also with high relative humidity values, the p(nBA/MMA) matrix is subject to scission of the polymeric chain by hydrolysis, which increases the presence of low molecular weight oxidation products [75]. Despite the hydrolytic degradation of these bonds, the single-shot method used during Py-GC/MS analyses did not allow to detect the presence of other compounds (additives) in small quantities within the polymeric structure of the acrylic binder. Even if this analysis process does not allow to obtain information on the contribution of the additives in the degradation process of the acrylic binder (data that could not have been related to a surface degradation such as the FTIR spectra, since the sample was analyzed in bulk), it allowed to better evaluate the acrylic matrix, and to understand which polymeric fractions were most affected by the oxidative as well as the hydrolytic reactions of ozone accelerated aging. Future studies and experiments will be carried out mainly considering the influence and behavior of additives, considering the study presented here as a preliminary evaluation of the qualitative and statistical results processed simultaneously by data fusion. Lastly, the comparison between the chromatograms of the unaged and aged samples mixed with the inorganic pigments (for both styrene-acrylic and acrylic binder) was complex, and a degradation pattern was not so evident. A more accurate and complete interpretation of these trends will be presented in the following statistical evaluations using data fusion.

### 3.2. Data Fusion Evaluation

#### 3.2.1. Styrene-Acrylics

The PCA model calculated on styrene-acrylics fused data retained a total of three PCs, accounting for 77.55% of explained variance. The corresponding PC1 and PC2 score plot is reported in Figure 6A, which shows that the samples aged at RH values equal to 50% are grouped into a separate cluster at positive score values of PC1 while the samples aged at RH values equal to 80% are separated from the unaged samples along PC2. It has to be highlighted that, in the PC1–PC2 score space, the samples aged at RH 50% are more distant from unaged samples, and the separation between unaged samples and samples aged at RH 80% is less pronounced. These findings suggest that for paint layers made of styrene-acrylic binder, the degradation phenomena occurring due to ozone exposure and relative humidity are more evident when RH is around 50%. The corresponding loadings are reported in Figure 6B,C for FTIR and Py-GC/MS variables, respectively. To simplify the interpretation of the results, the loadings obtained from the two data blocks were reported in separate figures. However, it has to be noted that the PCA model was calculated on the fused dataset. Therefore, it accounts for variation sources common to FTIR and Py-GC/MS data.

As discussed above, PC1 mainly accounts for the variations occurring in the samples aged in the presence of ozone and RH 50%, which have positive PC1 score values. Considering the PC1 loading vector of FTIR data reported (Figure 6B, in blue colour), the spectral regions with higher relevance on this PC are those falling in the 1800–1600 cm^−1^ region, suggesting that the spectra of the samples aged at RH 50% have higher intensity in this spectral region, mainly characterized by the formation of the peak at 1704 cm^−1^. On the other hand, PC2 describes the differences between unaged samples and RH 80% samples. The PC2 loadings of FTIR variables (Figure 6B) generally have positive loading values, with the more relevant spectral regions in the 3030–2800 cm^−1^ interval (C–H stretching) and in the peak at 1727 cm^−1^ (C=O stretching). This indicates that the RH 80% samples, which have positive PC2 score values, are characterized by increased spectra intensity in these regions. Considering the degradation trend shown by the samples aged at 50% and 80% RH, it is possible to confirm the evaluations presented in Section 3.1.1. In fact, in the first case, ozone is more reactive if in a humidified environment at 50%, causing the decomposition of the phenyl group and the splitting of the double bonds (shoulder formation at 1704 cm^−1^). At 80%, on the other hand, the main degradation reaction is the hydrolysis of binder bonds showing a greater intensity of the main functional groups than the unaged samples.

In the PC1 loadings of the Py-GC/MS block (Figure 6C), it has to be noticed that the majority of the variables have negative PC1 loading values. Therefore, the RH 50% samples have a generally lower abundance of the compounds identified by Py-GC/MS, compared to the unaged or RH 80% samples. However, the compounds that mainly show a different degradation trend for samples aged at 50% RH are nBA-styrene dimer 2 (SC26), styrene dimer 3 (SC31), and styrene trimer 2 (SC41). These results support the evaluations previously mentioned, i.e., ozone is able to rearrange itself in a double layer system and has more concentrated oxidative properties at 50% RH, mainly affecting the styrene fractions. Concerning the PC2 loadings of the compounds measured by Py-GC/MS, the more relevant variables characterizing RH 80% samples (i.e., those with higher positive PC2 loading values) are: ethylbenzene (SC3), benzene, (1-methylethyl) (SC5), n-butyl methacrylate (SC8), dibenzyl (SC22), and nBA-styrene dimer (SC24). These results suggest that the methacrylic fraction is mainly subject to hydrolysis reactions at 80% RH, as seen in the FTIR results. However, it is also possible to notice that, despite the fact that the oxidative power of ozone is not as high as at 50% RH, it equally attacks the benzene groups of the styrene component. This degrading trend was not particularly visible by FTIR analyses. Lastly, the influence that various inorganic pigments have on the stability of the polymeric binder is evident. From the simultaneous evaluation of the spectral contributions obtained by FTIR analysis and the organic fractions identified by Py-GC/MS, it is possible to observe that PW6 promotes a stronger degradation at 80% RH (positive values of PC2), while PY37 at 50% RH (negative values of PC2).

#### 3.2.2. Acrylics

Considering fused data of acrylic samples, the PCA model was calculated retaining two PCs, which account for 81.89% of the explained variance (Figure 7). The PC1 and PC2 score plot (Figure 7A) shows that the samples are grouped based on aging conditions. In particular, PC1 describes the differences between unaged and aged samples, while PC2 separates the aged samples according to the exposure to different RH values. In particular, aged samples (both RH 50% and 80%) are characterized by positive PC1 score values. However, RH 50% samples have negative PC2 score values, and RH 80% samples have positive PC2 score values. Therefore, in acrylics samples, the main source of variability is ascribable to degradation patterns due to ozone exposure, common to both RH 50% and RH 80% samples, and described by PC1. In addition, the combination of ozone exposure and different relative humidity conditions determine differences between RH50% and RH80% samples, according to PC2 score values.

Considering the PC1 loadings of the FTIR data block (Figure 7B), all the spectral variables have positive PC1 loading values, suggesting a general increase in the intensity of the spectra in aged samples. Therefore, PC1 describes a general degradation pattern common to RH 50% and 80% samples. Considering PC2 loadings of FTIR data (Figure 7B), which describe the differences between RH 50% and RH 80% samples, it is possible to observe that the modifications at RH50% determine a higher increase of intensity of the peak at 1726 cm^−1^ (C=O stretching) and the formation of a peak at 1650 cm^−1^, due to the formation of aldehydes, ketones, and carboxylic acids. Conversely, for RH 80% samples, the degradation phenomena mainly affect the peak at 2891 cm^−1^ (surfactant) with a general increase of intensity of this band. These results confirm the assessments related to the various degradation trends observed by FTIR spectra analysis.

Considering the Py-GC/MS data block, the variables with positive PC1 loading values are those with higher relative abundance in aged samples (Figure 7C). These variables correspond to the majority of the compounds belonging to the group “monomeric fractions”. This explains that, regardless of the percentage of relative humidity applied during artificial aging, the main monomer groups (i.e., MMA and BMA) are particularly subject to hydrolysis reactions. On the other hand, concerning PC2 loadings of the Py-GC/MS data block, variables with positive PC2 loading values are those with higher relative abundances in RH 80% samples, while the variables with negative PC2 loading values mainly characterize RH 50% samples. Given these considerations, the negative PC2 loading values of the Py-GC/MS data block, reported in Figure 7C, show that in RH 50% samples the more abundant compounds belonging to the group “solvents + monomeric fractions”, i.e., propene (AC1), butene (AC2), n-butyl aldehyde (AC3), methyl methacrylate MMA (AC8), butyl acrylate BA (AC12), and benzene (1-ethyl-2-propenyl) (AC17). On the other hand, RH 80% samples are well defined in the score plot by the high concentration of styrene (AC13), n-butyl methacrylate nBMA (AC14), n-butyl tiglate (AC16), and nBA-MMA dimer (AC20). As previously explained for the FTIR and PY-GC/MS results, the acrylate fractions are equally subject to the degradation processes by the interaction between ozone and humidity. Although pyrograms obtained without derivatization do not show a particular presence of additives, the global explained variance described in the score plot also considers the loading plots of the IR contribution in which the additive variable (2891 cm^−1^) was added. In this way, it was possible to differentiate the samples aged at 80% RH from those of 50% RH, and therefore to confirm that the migration of the surfactant to the surface is a relevant phenomenon in the investigation of the chemical stability of acrylic paints when exposed to ozone.

As for styrene-acrylic paints, unlike the Py-GC/MS characterization, it is possible to observe how the pigments influence the ozone degradation phenomena. In fact, observing the score plot (Figure 7A), for RH 50% samples, the pigment that most promotes degradation reactions is PR101, while for RH 80% samples, PB28. For more detailed and specific knowledge of the deteriorating contributions of pigments, a semi-quantitative evaluation of the FTIR results will be presented in the following chapter.

### 3.3. Semi-Quantitative FTIR Evaluation of Pigment Influence in the Degradation Process

As demonstrated in several studies [76,77,78], the contribution of pigments in the degradation process of polymeric films is a factor to be considered since their chemical composition, crystalline properties, and chemical-physical features can promote or inhibit deteriorating reactions deriving from atmospheric pollutants. According to the chosen aged treatment, two different integration peaks were considered during the semi-quantitative evaluation of the IR spectra. This is due to the fact that different degradation products are obtained at different RH values combined with ozone. Specifically, the peak at 1703 cm^−1^ was integrated in the spectra of samples aged at 50% RH, while the peak at 1650 cm^−1^ was integrated for those aged at 80% RH (Figure 8a). Both spectral signals represent the formation of unsaturated ketones and aldehydes after the subsequent breaking of the double bonds of the phenyl groups. However, it is interesting to note how the inorganic pigments in the mixture favor this reaction differently according to the various concentrations of relative humidity. Specifically, considering that the paints undergo a strong superficial degradation already at 50% RH, those mixed with PR101, PY37, and PB29 seem to increase this process. On the other hand, paints mixed with PW6, PB35, and PB28 increase the formation of carboxylic acids when the humidified water component is more significant (80% RH).

As discussed for styrene-acrylic paints, the inorganic pigment component can, in some cases, further compromise the stability of the organic binder when exposed to gaseous pollutants (Figure 8b). Therefore, to evaluate its effects for the acrylic paints aged at 80%, the band at 1343 cm^−1^ (PEO signal) was integrated, while for those aged at 50%, the band at 1650 cm^−1^ deriving from the degradation of the acrylic component. The semi-quantitative evaluation confirms that PW6, PB28, and PB35 better favor the migration of the surfactant when the quantity of water is high (80%), while PY37, PR101, and PB29 promote the degradation of the organic component. This evaluation, combined with the results obtained by the data fusion, confirms the different influences of the pigments on the stability of the binders under examination and the importance of a simultaneous evaluation of results for an optimized interpretation of the degradation trends.

## 4. Conclusions

This study performed accelerated ozone aging in a humidified environment (50% and 80% RH) on modern painting samples. The considered polymeric binders are two emulsions commonly used in the artistic field (acrylic and styrene-acrylic-based formulations) mixed with various inorganic pigments. After aging, the samples were investigated by means of ATR-FTIR spectroscopy and Py-GC/MS analysis. The data collected from the two analytical techniques were jointly evaluated using a low-level data fusion approach combined with principal component analysis (PCA). FTIR spectroscopy and Py-GC/MS chromatography can be considered two complementary analytical techniques since FTIR measurements acquire information about the sample surface, while Py-GC/MS determines the whole sample composition (bulk analysis). Given these considerations, the evaluation of both datasets performed with data fusion allowed to compare the chemical information deriving from the various constituent materials of the samples and their aging variables.

From the qualitative ATR-FTIR and Py-GC/MS analysis, the various binders showed different results depending on the aging conditions. The styrene-acrylic emulsion is more subject to the deteriorating action of ozone at 50% RH. In fact, the hydrolysis reactions of all polymeric matrices, the breakdown of saturated aliphatic chains (mainly affecting the phenyl groups of styrene component), and the formation of degradation products (band at 1703 cm^−1^) are the main degradation factors observed. On the other hand, the acrylic emulsion shows a double degradation behavior by ATR-FTIR analysis. With both 50% and 80% RH, the hydrolysis reactions are shown (due to the water action film). However, at 80% RH, an increase in the spectral intensity of the surfactant functional groups is observed (due to its surface migration when exposed to high humidity concentrations). This last observation could not be confirmed by the Py-GC/MS analyses (due to the selected measurement mode). Being able to evaluate the acrylic matrix more specifically, it was evident that the p(nBA/MMA) component is also subject to scission of the polymeric chain by hydrolysis at 80% RH.

The employed data fusion approach confirmed the previously obtained results and brought further information about the aging variables that are most relevant in the degradation processes, which polymeric fractions were most affected by ozone aging, and which inorganic pigments promoted these deteriorating behaviors. Specifically, for styrene-acrylic paints, ozone is more reactive at 50% RH, causing the decomposition of the phenyl group and the splitting of the double bonds (mainly affecting nBA-styrene dimer, styrene dimer, and styrene trimer), while at 80% RH, hydrolysis is the main bond degradation reaction (mainly affecting the methacrylic fractions). Furthermore, by the simultaneous evaluation of FTIR and Py-GC/MS results, PW6 promotes degradation at 80% RH, while PY37 at 50% RH. On the other hand, acrylic paints show a common degradation trend at both RH 50% and RH 80%. In the first case, RH50% determines the hydrolysis of the carboxylic group (mainly n-butyl aldehyde, MMA, and BA fractions) and the formation of degradation products, while at RH 80%, the degradation phenomena mainly affect the migration of the surfactant to the surface (Py-GC/MS analysis shows degradation of the nBMA and nBA-MMA dimer fractions). Moreover, in this case, the simultaneous evaluation of the results shows that PB28 promotes more degradation at 80% RH, while PR101 at 50% RH. The semi-quantitative FTIR evaluation confirmed the results related to the influence of pigments on the binder stability.

This study provides important outcomes. Despite the fact that artworks can be exhibited in different exhibition environments, pollutants inevitably affect the artistic materials. Therefore, understanding the still unclear degradation reactions is of fundamental importance for their preservation and conservation. This study highlighted how the results obtained from techniques commonly used in the cultural heritage field (ATR-FTIR and Py-GC/MS) can be implemented by the use of multivariate methods (data fusion and PCA) able to extract further information on the chemical stability of artistic materials and allowing to optimize the conservation and restoration practices of modern and contemporary artworks. The use of data fusion represents a preliminary approach for future research studies and applications, both extensive in the cultural heritage sector and innovative in other fields (industrial, pharmaceutical, biology, coatings, sensors, etc.). Relevant topics for future studies may include the improvement of instrumental parameters to obtain more accurate or specific results (such as the detection of surfactant fractions by Py-GC/MS), the cross-evaluation of multiple pollutants, a comparison with accelerated aging tests and real aged samples, and the use of additional techniques in the simultaneous statistical evaluation. These further considerations will provide additional information on the stability of polymeric films and the oxidizing effect that pollutants have on various art materials.

## Figures and Tables

**Figure 1 polymers-14-01787-f001:**
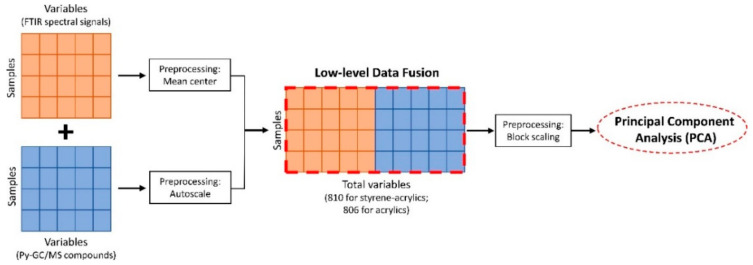
Schematic representation of the Data Fusion workflow used to evaluate FTIR and Py-GC/MS data obtained.

**Figure 2 polymers-14-01787-f002:**
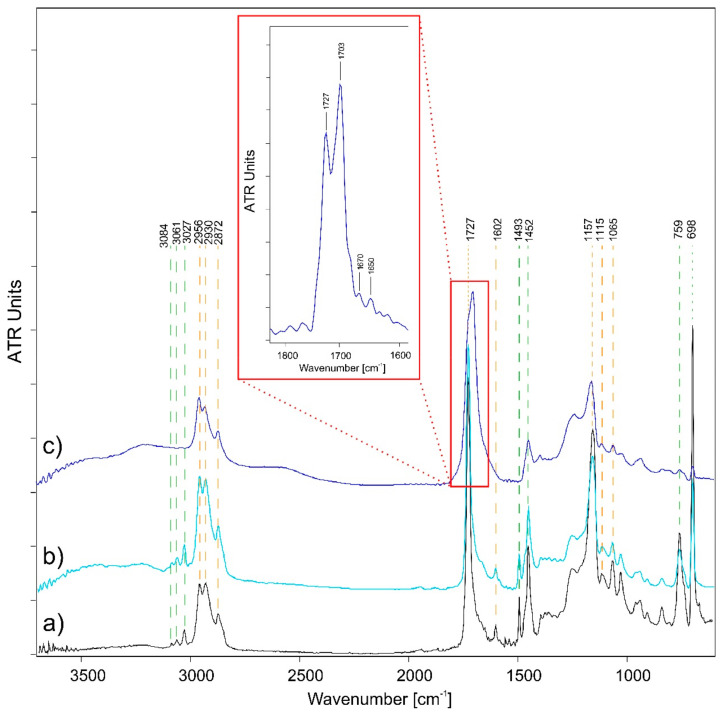
ATR-FTIR spectra comparison of pure styrene-acrylic films: (**a**) unaged (black), (**b**) 80% RH (light blue), and (**c**) 50% RH (blue) aged samples are showed.

**Figure 3 polymers-14-01787-f003:**
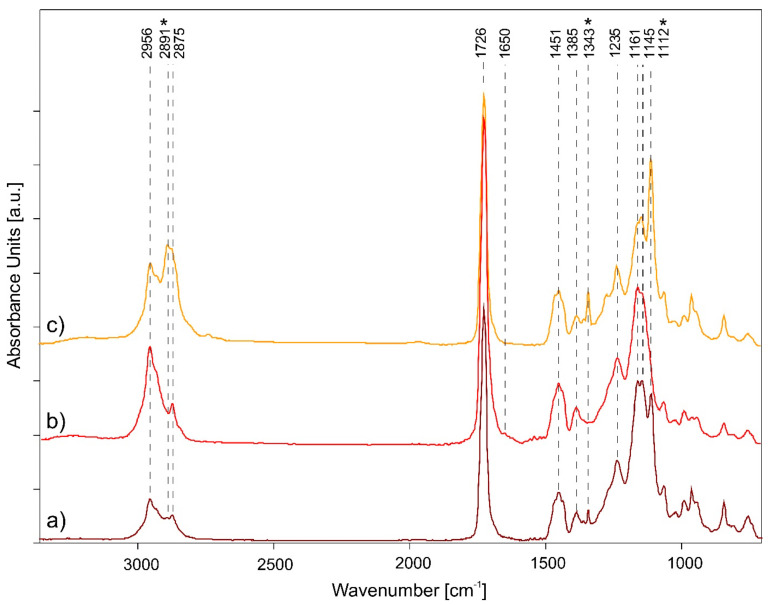
ATR-FTIR spectra comparison of pure acrylic films: (**a**) unaged (dark red), (**b**) 50% RH (red), and (**c**) 80% RH (yellow) aged samples are shown. The three main spectral signals of PEO surfactant are signed with *.

**Figure 4 polymers-14-01787-f004:**
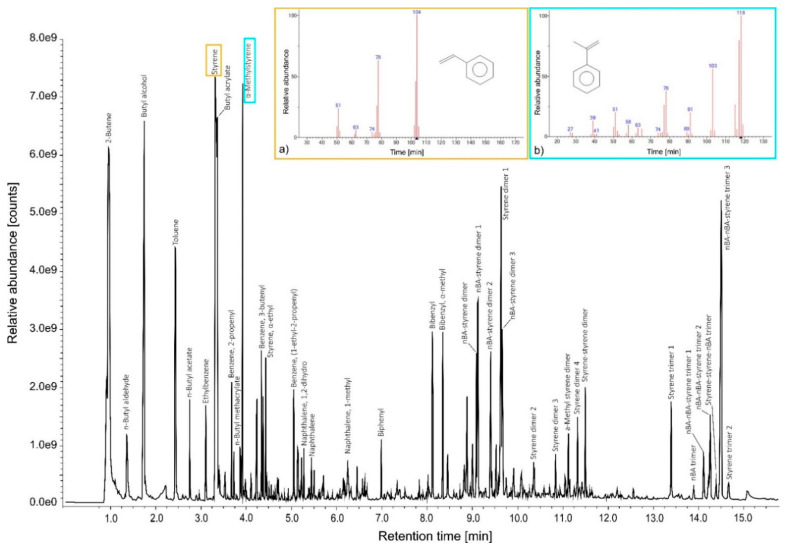
TIC (total ion current) pyrogram obtained after Py-GC/MS analysis of the pure unaged styrene-acrylic emulsion fragments. (**a**) Styrene; (**b**) α-Methylstyrene. Mass spectra of the highest intensity peaks detected are illustrated. All identified compounds are listed in Table 2.

**Figure 5 polymers-14-01787-f005:**
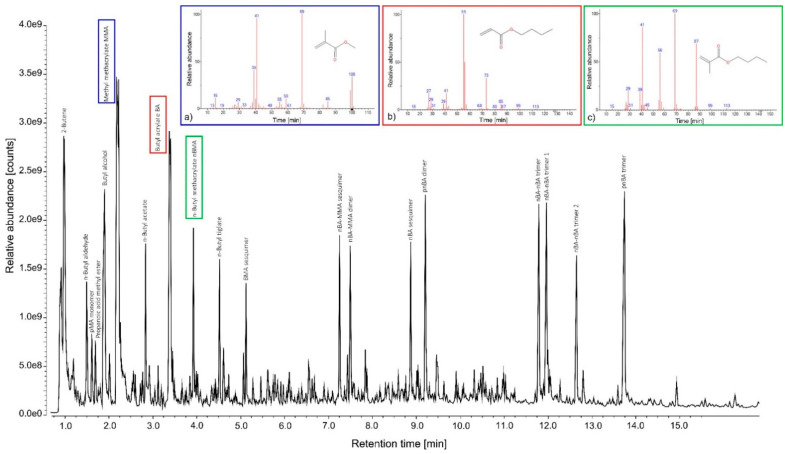
TIC (total ion current) pyrogram obtained after Py-GC/MS analysis of the pure unaged acrylic emulsion fragments. (**a**) Methyl methacrylate MMA; (**b**) Butyl acrylate BA; (**c**) n-Butyl methacrylate nBMA. Mass spectra of the highest intensity peaks detected are illustrated. All identified compounds are listed in Table 3.

**Figure 6 polymers-14-01787-f006:**
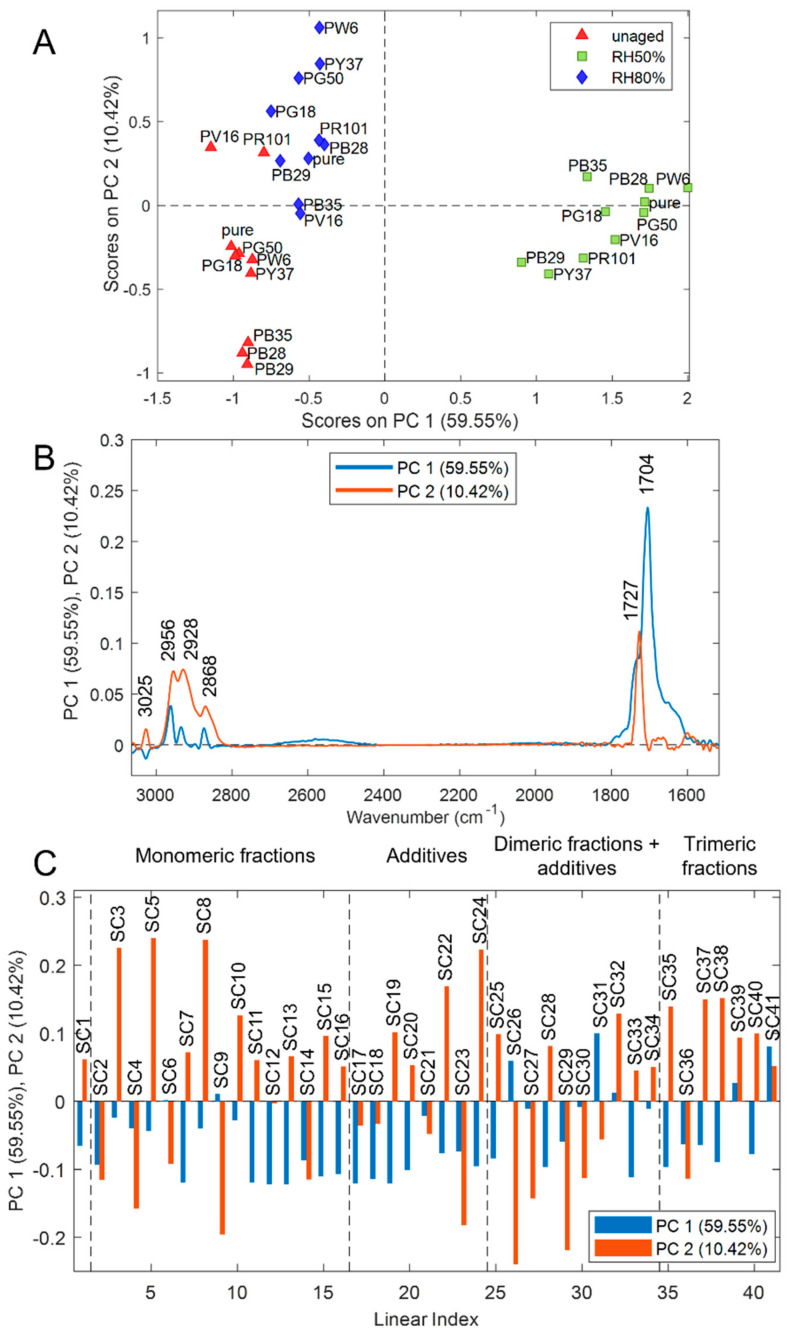
Results of PCA model calculated on FT−, IR and Py−, GC/MS fused dataset of styrene-acrylic samples: (**A**) PC1 and PC2 score plot, (**B**) PC1 and PC2 loadings of FT-IR variables, and (**C**) PC1 and PC2 loadings of Py-GC/MS variables.

**Figure 7 polymers-14-01787-f007:**
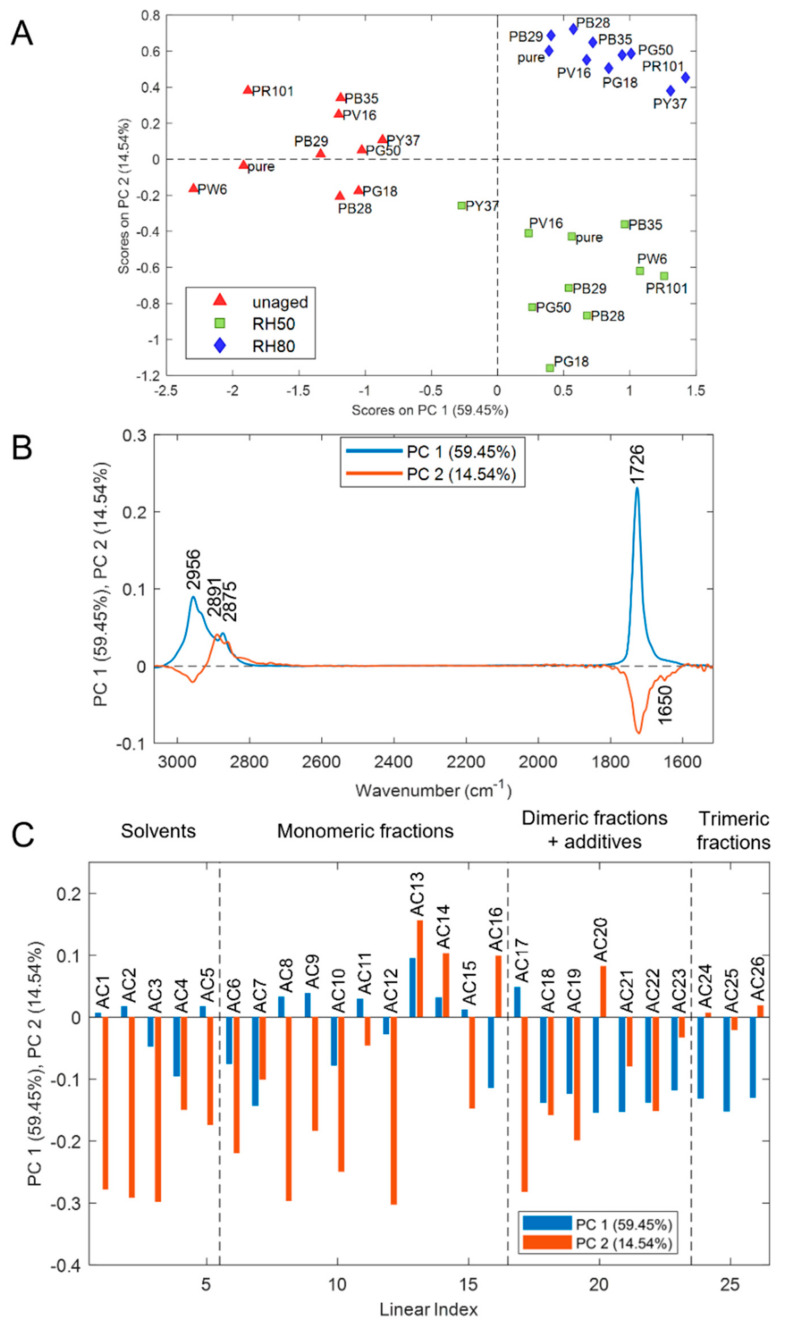
Results of PCA model calculated on FT−, R and Py−, GC/MS fused dataset of acrylic samples: (**A**) PC1 and PC2 score plot, (**B**) PC1 and PC2 loadings of FT-IR variables, and (**C**) PC1 and PC2 loadings of Py-GC/MS variables.

**Figure 8 polymers-14-01787-f008:**
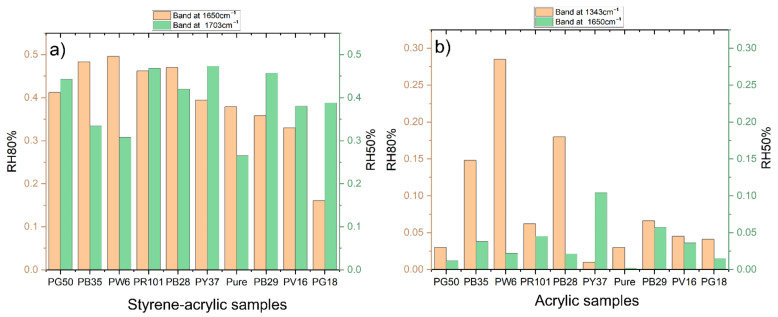
Semi-quantification evaluation of selected spectral signals on (**a**) styrene-acrylic and (**b**) acrylic paints. Comparison between integration areas of 50% RH aged (green) and 80% RH aged (orange) samples.

**Table 1 polymers-14-01787-t001:** List of materials analyzed.

**Pigment Name**	**Chemical Composition**	**Colour Index (C.I.) Number**
Titanium white	TiO_2_	PW6
Cadmium yellow	CdS	PY37
Cobalt green	Co_2_TiO_4_	PG50
Hydrated chromium oxide green	Cr_2_O_3_ · 2H_2_O	PG18
Cobalt blue	CoO · Al_2_O_3_	PB28
Cerulean blue	CoSnO_3_	PB35
Artificial ultramarine blue	Na_8–10_Al_6_Si_6_O_29_S_2–4_	PB29
Iron oxide red	Fe_2_O_3_	PR101
Manganese violet	NH_4_MnP_2_O_7_	PV16
**Binder Name**	**Chemical Composition**	**Commercial Name**
Acrylic emulsion	p(nBA/MMA)	Plextol^®^ D498
Styrene-acrylic emulsion	Styrene acrylate co-polymer	Acronal^®^ S790

**Table 2 polymers-14-01787-t002:** Compounds identified in unaged pure styrene-acrylic emulsion by Py-GC/MS analysis. The most intense peaks were listed in bold.

Sample Name	RT (min)	Compounds	*m*/*z*
Styrene-acrylic emulsion	0.97	2-Butene	41 (56)
	1.40	n-Butyl aldehyde	44 (27, 72)
	1.79	Butyl alcohol	56 (31)
	2.46	Toluene	91
	2.78	n-Butyl acetate	43 (56)
	3.13	Ethylbenzene	91 (106)
	3.33	**Styrene**	104 (78)
	3.37	**Butyl acrylate BA**	55 (73)
	3.69	Benzene, 2-propenyl-	117 (91)
	3.88	**n-Butyl methacrylate nBMA**	69 (41, 87, 56)
	3.93	**α-Methylstyrene**	118 (103, 78)
	4.35	Benzene, 3-butenyl-	91 (132)
	4.44	Styrene, α-ethyl-	117 (132)
	5.05	Benzene, (1-ethyl-2-propenyl)-	118
	5.15	Benzene, 3-butynyl-	91 (129)
	5.28	Naphthalene, 1,2-dihydro-	130 (129, 115)
	5.45	Naphthalene	128 (127, 102)
	6.25	Naphthalene, 1-methyl-	142 (141, 115)
	6.98	Biphenyl	154 (153, 152)
	8.11	Bibenzyl	91 (182, 65)
	8.33	Bibenzyl, α-methyl-	105 (104, 77)
	9.07	nBA-styrene dimer	91 (115)
	9.11	nBA-styrene dimer 1	91 (104)
	9.38	nBA-styrene dimer 2	91 (130)
	9.41	pnBA dimer	127 (98)
	9.62	Styrene dimer 1	91 (130, 104)
	9.64	nBA-styrene dimer 3	131 (91, 115)
	10.35	Styrene dimer 2	117 (91, 115)
	10.81	Styrene dimer 3	130 (115, 91)
	11.09	a-Methyl styrene dimer	118 (91, 117)
	11.30	Styrene dimer 4	143 (142, 128)
	11.47	Styrene-styrene dimer	142 (129, 115)
	13.35	Styrene trimer 1	91 (128, 171)
	13.84	nBA trimer	41 (57, 134)
	14.07	nBA-nBA-styrene trimer 1	98 (91, 126)
	14.22	nBA-nBA-styrene trimer 2	212 (142, 118)
	14.34	Styrene-styrene-nBA trimer	91 (117)
	14.47	nBA-nBA-styrene trimer 3	131 (91, 117)
	14.62	Styrene trimer 2	91 (117, 115)

**Table 3 polymers-14-01787-t003:** Compounds identified in unaged pure acrylic emulsion by Py-GC/MS analysis. The most intense peaks were listed in bold.

Sample Name	RT (min)	Compounds	*m*/*z*
Acrylic emulsion	0.97	2-Butene	41 (28, 56)
	1.40	n-Butyl aldehyde	44 (27, 72)
	1.48	Acrylic acid methyl ester	55 (27, 85)
	1.56	Propanoic acid methyl ester	57 (29, 88)
	1.79	Butyl alcohol	56 (31)
	2.07	**Methyl methacrylate MMA**	69 (41, 100)
	2.78	n-Butyl acetate	43 (56)
	3.37	**Butyl acrylate BA**	55 (73)
	3.89	**n-Butyl methacrylate nBMA**	69 (41, 87, 56)
	4.47	n-Butyl tiglate	83 (55, 101)
	5.54	BMA sesquimer	121 (126, 136, 67)
	7.30	nBA-MMA sesquimer	143 (129, 83)
	7.53	nBA-MMA dimer	112 (67, 95)
	9.07	nBA sesquimer	115 (87, 171)
	9.41	p*n*BA dimer	127 (126, 98)
	12.06	nBA-nBA trimer	195 (93, 194)
	12.22	nBA-nBA trimer 1	195 (93, 194, 250)
	12.85	nBA-nBA trimer 2	149 (148, 121)
	13.86	p*n*BA trimer	181 (134, 236)

## Data Availability

Additional FTIR spectra, Py-GC/MS chromatograms, information concerning the data fusion approach, or semi-quantitative evaluations are available upon request.

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
