# Peer review of "Data Fusion Approach to Simultaneously Evaluate the Degradation Process Caused by Ozone and Humidity on Modern Paint Materials"

_polymers, 2022, doi:10.3390/polym14091787_

Round 1
Reviewer 1 Report
The paper entitled “Data Fusion approach to simultaneously evaluate the degradation process caused by ozone and humidity on modern paint materials” reported an interesting work on the very important topic of degradation processes of painted materials caused by environmental factors. The article is well written, summarizes a lot of valuable information in its figures and tables, and contains a lot of work. The manuscript needs several revisions before it can be published. Therefore, please improve or clarify the following points:
- It is not clear how the mitigation of surfactant to the surface is investigated. You must give more details in the last paragraph of the introduction on page 3.
- The resolution of Figures 1, 4, 5 and A1 is low. Please improve it.
- The homogeneity of the reference section needs to be maintained. Some references are abbreviated, while others are not. Please check and revise the whole reference section according to the Polymers instruction for authors (https://www.mdpi.com/journal/polymers/instructions )
Based on these, I advise the authors to rectify the above mentioned issues, and I hope to re-evaluate the revised manuscript.
Reviewer 2 Report
In the present study, the authors have developed data fusion approach to examine the degradation of acryl-based paints. Painting samples mixed with various inorganic pigments were degraded by accelerated ozone aging and investigated by ATR-FTIR and Py-GC/MS. The data were jointly evaluated by principal components analysis (PCA). I find it interesting to try applying PCA to degradation analysis. The present method has succeeded in showing the difference in deterioration behavior depending on the type of base polymer binders. The interpretation of PC1 and PC2 needs to be refined, but I think that is a topic for the future. I think this paper deserves to be published in Polymers almost as it is.
Please check the following:
-- page 5, line 17 from the bottom: "(0.000mg)" should be replaced with a numerical value other than 0.000.
-- page 12, line 7: "identified" should be "identify."
Reviewer 3 Report
This study deals with the analytical and statistical investigations of the degradation processes of acrylic and styrene-acrylic paints after exposure to ozone (O3) and RH. The authors used different pigments used in murals and paintings, investigating them by FTIR and GC-MS.
The paper is well structured, detailed accordingly, well-argued, and very useful for readers. No further comments from my side.
I appreciate that this paper could be published in the present version.
